# Collaborative Decision Making
# Using Action Suggestions

**Dylan M. Asmar**
Stanford Intelligent Systems Laboratory
Stanford University
Stanford, CA 94305
asmar@stanford.edu

**Mykel J. Kochenderfer**
Stanford Intelligent Systems Laboratory
Stanford University
Stanford, CA 94305
mykel@stanford.edu

## Abstract

The level of autonomy is increasing in systems spanning multiple domains, but these systems still experience failures. One way to mitigate the risk of failures is to integrate human oversight of the autonomous systems and rely on the human to take control when the autonomy fails. In this work, we formulate a method of collaborative decision making through action suggestions that improves action selection without taking control of the system. Our approach uses each suggestion efficiently by incorporating the implicit information shared through suggestions to modify the agent's belief and achieves better performance with fewer suggestions than naively following the suggested actions. We assume collaborative agents share the same objective and communicate through valid actions. By assuming the suggested action is dependent only on the state, we can incorporate the suggested action as an independent observation of the environment. The assumption of a collaborative environment enables us to use the agent's policy to estimate the distribution over action suggestions. We propose two methods that use suggested actions and demonstrate the approach through simulated experiments. The proposed methodology results in increased performance while also being robust to suboptimal suggestions.[1]

## 1   Introduction

Autonomous systems and humans have different strengths. Moravec [1] claimed in 1988 that "It is comparatively easy to make computers exhibit adult level performance on intelligence tests or playing checkers, and difficult or impossible to give them the skills of a one-year-old when it comes to perception and mobility." Thirty years later, computers have achieved superhuman level performance on many games like Go and StarCraft II [2], [3]. Despite major strides in perception, algorithms still exhibit brittleness and struggle with consistency in tasks that most would consider trivial for a young adult, like reliably generating a plausible sentence from a list of words [4]–[7].

Humans often make suboptimal decisions; however, they are exceptionally good at finding ways to accomplish tasks, balance risk, reason through unstructured problems, and incorporate new information into decisions without prior experience in a situation. Systems integrating autonomy often take advantage of human strengths by using a human-on-the-loop structure. This framework allows for humans to use their experience to recognize situations where the autonomy might struggle or is failing and to take control [8], [9]. We hypothesize that instead of requiring the human to assume control, an agent can use occasional control inputs from the human-on-the-loop in the form of action suggestions to improve its understanding of the environment.

---

[1]Code is available at https://github.com/sisl/action_suggestions.

36th Conference on Neural Information Processing Systems (NeurIPS 2022).

In this work, we explore a method of collaboration that enables combining benefits of heterogeneous systems like humans and machines through action suggestions versus direct control. Our approach was inspired by the way that we see humans incorporate action suggestions in daily life. A real world example is a pilot flying an aircraft while coordinating with air traffic control (ATC). In the United States, there are various services ATC provides to ensure safety. Many of these services provide suggestions to pilots, but do not mandate an action or remove responsibilities from the pilot (e.g. radar assistance to VFR aircraft) [10]. Consider a situation where a pilot is flying at low altitude, has knowledge of another aircraft to the left, and an ATC controller recommends a turn to the left. This action suggestion would cause the pilot to consider situations where a left turn towards traffic would be optimal (e.g. terrain ahead) and reassess their belief about the environment.

In our problem of interest, we are assuming the agent has the same objective as a suggester and use action suggestions as additional information about the environment to increase the quality of the action selection. We avoid having to share exact belief states between agents by treating the suggested action as an observation. Using the assumption that the suggester is collaborative, we use the agent's policy to infer a distribution over suggested actions and use this distribution to update the agent's belief. This approach allows collaboration between different types of systems and avoids explicit translation of complex belief spaces.

In this paper, we first review related work in section 2 and discuss how our approach differs from previous methods. Our primary contributions are in section 3. We first provide an overview of a partially observable Markov decision process (POMDP) and then outline the problem setting. Section 3.3 demonstrates how we can view action suggestions as independent observations and modify the belief update process. We then provide two methods to estimate the distribution over suggested actions using the agent's policy in order to update the belief in section 3.4. Section 4 presents the experiments and discusses key results demonstrating the effectiveness of our approach.

## 2 Previous Work

Collaborative sequential decision making between two agents is a form of shared autonomy. The exact definition of shared autonomy and control varies in the literature [11]. A common theme among the approaches is combining inputs between two agents to achieve performance better than the agent's acting alone. The problems often involve one of two main categories: 1) assistance from an autonomous agent to a human or 2) assistance from a human to the autonomous system. The problem framework we are considering aligns with the second category.

In the first category, an autonomous agent is seeking to perform actions to assist the other agent (often a human). Dragan and Srinivasa [12] discuss key aspects of this category and decompose it into prediction of the user's intent and arbitration between the user's inputs and the action chosen by the autonomous system. Nguyen *et al.* [13] propose a method that decomposes a game into subtasks that can be modelled as a Markov decision process. The autonomous agent then infers the subtask the other agent is attempting to accomplish and selects actions to assist. Others model this category as a POMDP where the uncertainty is around the goal of the assisted agent [14]–[16].

The second category often involves forms of corrections. Losey and O'Malley [17] use aspects of optimal control to learn from physical human interventions. Nemec *et al.* [18] first learn from human demonstrations and then refine the policy through incremental learning from kinesthetic guidance of the autonomous agent. Other approaches perform the corrections in an iterative cycle during a planning process. Reardon *et al.* [19] present a method that allows a human to interactively provide feedback during planning. The feedback is in the form of hints or suggestions and is used to modify the optimization process. This approach is similar to the concept of reward shaping in reinforcement learning.

Cognetti *et al.* [20] provide a method that allows for real time modifications of a path while Hagenow *et al.* [21] present a method that permits an outside agent to modify key robot state variables. These changes are then blended with the original control. The concept of following guidance versus acting independently and how to blend the two was explored by Evrard and Kheddar [22] where they use two controllers and alternate roles of following and leading. Medina *et al.* [23] present a method that selects between two strategies depending on the level of disagreement between agents.

Another related area of research is imitation learning. Within this field, there are methods to learn from feedback by querying the expert during training or expert intervention real-time [24], [25]. These methodologies are a form of human-machine collaboration by using the expert knowledge of the human to train or correct the system and then letting the system perform the task autonomously. These approaches generally rely on strict assumptions or an explicit model of the other agent so the autonomous system can interpret inputs in a way to reason how to integrate them. In the first category, Jeon *et al.* [26] propose a model structure to map human inputs to different actions based on the robot's confidence in the goal. Reddy *et al.* [27] deviate from the use of model based methods by using human-in-the-loop deep reinforcement learning to map from observations and user inputs to an agent action.

Our approach builds on the concept of mapping user inputs to an agent action. However, we differ from other methods in that we do not build an explicit model of the suggester nor learn a mapping that might be suggester dependent. We achieve this mapping by assuming the suggester is collaborative and shares the same objective when providing inputs. This assumption enables the use of the key insight from Spencer *et al.* [25], "... any amount of expert feedback ... provides information about the quality of the current state, the quality of the action, or both", and we build an implicit model of the suggester using only the agent's knowledge of the environment and the agent's policy.

## 3  Action Suggestions as Observations

### 3.1  Background

A partially observable Markov decision process (POMDP) is a mathematical framework to model sequential decision making problems under uncertainty [28]. A POMDP is represented as a tuple $(\mathcal{S}, \mathcal{A}, \mathcal{O}, T, O, R, \gamma)$, where $\mathcal{S}$ is a set of states, $\mathcal{A}$ is a set of actions, and $\mathcal{O}$ is a set of observations. At each time step, an agent starts in state $s \in \mathcal{S}$ and chooses an action, $a \in \mathcal{A}$. The agent transitions from state $s$ to state $s'$ based on the transition function $T(s, a, s') = p(s' \mid s, a)$, which represents the conditional probability of transitioning to state $s'$ from state $s$ after choosing action $a$. The agent does not directly observe the state, but receives an observation $o \in \mathcal{O}$ based on the observation function $O(s', a, o) = p(o \mid s', a)$, which represents the conditional probability of observing observation $o$ given the agent chose action $a$ and transitioned to state $s'$.

At each time step the agent receives a reward, $R(s, a) \in \mathbb{R}$ for choosing action $a$ from state $s$. For infinite horizon POMDPs, the discount factor $\gamma \in [0, 1)$ is applied to the reward at each time step. The goal of an agent is to maximize the total expected reward $\mathbb{E}\left[\sum_{t=0}^{\infty} \gamma^t R\left(s_t, a_t\right)\right]$, where $s_t$ and $a_t$ are the state and action at time $t$. One method to solve a POMDP is to infer a belief distribution $b \in \mathcal{B}$ over $\mathcal{S}$ and then solve for a policy $\pi$ that maps the belief to an action where $\mathcal{B}$ is the set of beliefs over $\mathcal{S}$ [29]. Executing with this type of policy requires maintaining $b$ through updates after each time step.

Policies can be generated offline or computed online during execution. In this work, we focus on applying our method to policies generated offline and leave the application to online solvers for future work. Many approximate offline solvers involve point-based value iteration. The idea is to sample the belief space and perform backup operations on the sampled points of the belief space, iteratively applying the Bellman equation until the value function converges. PBVI [30], HSVI2 [31], FSVI [32], and SARSOP [33] are examples of such an approach, though they differ in the selection of initial belief points, the generation of points at each iteration, and the choice of which points to backup. These algorithms represent the policy as a set of alpha vectors. In this work, we used SARSOP to generate the policies; however, any algorithm that produces a policy where the utility of a belief can be calculated could be implemented with little change to our methods.

### 3.2  Problem Formulation

For a given problem of sequential decision making under uncertainty, we choose to model the problem as a POMDP and use a point-based solution method. In this work, we assume discrete state, action, and observation spaces but the methods can be generalized to continuous spaces. Our problem involves two entities: an autonomous system using the POMDP policy to perform actions and interact with the environment that we will refer to as the agent, and a suggester providing action suggestions to the agent. The suggester can observe the environment, but not interact or affect the

state except to provide recommended actions to the agent. The suggester is not required to provide suggestions but can provide a maximum of one action suggestion at each time step.

The agent and suggester are collaborative and share the same objective of maximizing the total expected reward. The agent and suggester can receive different information and maintain separate beliefs of the environment. The separation of the actors allows each to process and receive information independently and capitalize on strengths differently. An example problem is an expert human receiving observations not modeled by an autonomous robot and providing intermittent suggestions.

### 3.3 Incorporating Action Suggestions

There are different ways for the agent to use action suggestions. One approach would be to model the suggester in the original POMDP. This approach is similar to previous shared autonomy work (section 2). Modeling the suggester within the POMDP would provide a fundamental way of incorporating suggestions, but would require a priori knowledge of the suggester to develop an explicit model to solve for the policy and also require maintaining a belief over the model during execution. A simple method that does not require a model of the suggester would be for the agent to naively follow each suggestion. A similar and more robust method would be to follow a suggestion if it meets some defined criteria, thus potentially disregarding suboptimal suggestions (e.g. only follow a suggestion if it is a specific action). These approaches are simple and can incorporate suggestions; however, they do not benefit from incorporating implied information contained with each suggested action.

Applying our inspirations we introduced in the flight example to our problem, each action suggestion contains information related to the suggester's belief of the environment. We propose treating each action suggestion as an observation of the state in order to update the agent's belief. This idea enables the suggester to influence the agent while the agent remains autonomous. Our problem assumes the suggester and the agent maintain independent beliefs. If we further assume the suggested action is not influenced by the agent's previous action, the suggested action is only dependent on the state. The independence of the agent's previous action and the suggester allows a modification of our belief update process that only involves the probability of receiving the suggested action given the current state. The suggested action is not always independent of the agent's action, but the assumption that it only depends on the current state is often reasonable.

Our belief at time $t$ over state $s \in \mathcal{S}$ with observations $o \in \mathcal{O}$ and action suggestions $o^s \in \mathcal{A}$ is $p(s_t \mid a_{0:t-1}, o_{o:t}, o^s_{o:t})$. Using Bayes' theorem, we can rewrite our expression as

$$p(s_t \mid a_{0:t^-}, o_{0:t}, o^s_{0:t}) \propto p(o^s_t \mid s_t, a_{0:t^-}, o_{0:t^-}, o^s_{0:t^-}) p(o_t \mid s_t, a_{0:t^-}, o_{0:t^-}, o^s_{0:t})$$
$$p(s_t \mid a_{0:t^-}, o_{0:t^-}, o^s_{0:t^-}) \quad (1)$$

where the subscript $0:t$ refers to all instances of that variable from $0$ to $t$, and $t^- = t - 1$. This expression can be simplified using the independence assumption, the law of total probability, and the Markov property to

$$p(s_t \mid a_{0:t^-}, o_{0:t}, o^s_{0:t}) \propto p(o^s_t \mid s_t) p(o_t \mid s_t, a_{t^-}) \sum_{s_{t^-} \in \mathcal{S}} p(s_t \mid s_{t^-}, a_{t^-}) p(s_{t^-} \mid a_{t^-}, o_{t^-}, o^s_{t^-}).$$
$$(2)$$

Equation (2) is a simple modification to our standard belief update procedure with POMDPs and is an expression of updating a belief with two independent observations.

### 3.4 Inferring the Distribution over Suggested Actions

We can update the agent's belief based on $p(o^s_t \mid s_t)$. However, the agent cannot calculate this distribution directly. We propose using the assumption that the agents are collaborative and estimate $p(o^s_t \mid s_t)$ by using the agent's policy. In our flight example, when the pilot receives the suggestion to turn left they would use their experience to reason through what scenarios they would have provided that same suggestion.

**Scaled Rational.** One approach is to assume the suggester is perfectly rational and has a policy identical to the agent. The suggester would only give actions that would maximize the total expected

reward using the agent's policy $\pi$. In other words, $p(o_t^s \mid s_t) \approx \mathbf{1}(o_t^s = \pi(s_t))$ where $\mathbf{1}(\cdot)$ is the indicator function. To relax the assumption of a perfectly rational agent and an identical policy, we introduce a scaling factor, $\tau \in (0, 1]$. With a scaling factor, we assume the suggester acts rationally and with the agent's policy a fraction $\tau$ of the time and uses a random policy otherwise. The scaled rational update can be expressed as

$$p(o_t^s \mid s_t) \approx \begin{cases} \tau, & \text{if } o_t^s = \pi(s_t) \\ \frac{1-\tau}{|\mathcal{A}|-1}, & \text{otherwise.} \end{cases} \tag{3}$$

**Noisy Rational.** Another approach is to assume the likelihood of the suggested action is related to the total expected reward of choosing that action. Shepard [34] developed a noisy rational model from a psychological perspective, and this model has been widely used in robotics to model suboptimal decision making [35]–[37]. With this model, the suggester is most likely to choose the action with the highest expected return and less likely to choose suboptimal actions. This model requires calculating the total expected return of each action.

The action value function $Q(s, a)$ returns the expected value of performing action $a$ in state $s$ and then executing optimally thereafter. We can use the agent's policy and the reward function and perform a one step look ahead to calculate the Q-function [29]. Using a policy represented with alpha vectors, we can calculate $Q(s, a)$ by performing a one step look ahead using a belief that the agent is in state $s$. The noisy rational model has the same form as the Boltzmann distribution and the softmax function. We can express this model as

$$p(o_t^s \mid s_t) \approx \frac{\exp\left(\lambda Q(s_t, o_t^s)\right)}{\sum_{a \in \mathcal{A}} \exp\left(\lambda Q(s_t, a)\right)} \tag{4}$$

where $\lambda \in [0, \infty)$ is a hyperparameter often referred to as the rationality coefficient. The distribution approaches a uniform distribution as $\lambda$ approaches 0. As $\lambda$ increases, the model approaches a perfectly rational agent.

## 4 Experiments

The proposed methodology to incorporate action suggestions as observations was evaluated on two classic POMDP problems, Tag [30] and RockSample [38]. These domains are relatively simple but provide a way to evaluate the merits of the approach by removing domain-specific variables that might influence performance. The simulations were constructed to first evaluate the effectiveness and efficiency of the proposed approach and then to test the robustness to suboptimal action suggestions.

### 4.1 Environments

The two environments chosen to evaluate our approach have discrete action, state, and observation spaces. The structure of the problems allow for the visualization of the belief space while also providing a modest scalability problem.

**Tag.** The Tag environment was first introduced by Pineau *et al.* [30]. The layout of the environment can be seen in fig. 1. The agent and an opponent are initialized randomly in the grid. The goal of the agent is to tag the opponent by performing the *tag* action while in the same square as the opponent. The agent can move in the four cardinal directions or perform the *tag* action. The movement of the agent is deterministic based on its selected action. A reward of $-1$ is imposed for each motion action and the *tag* action results in a $+10$ for a successful tag and $-10$ otherwise. The agent's position is fully observable but the opponent's position is unobserved unless both actors are in the same cell. The opponent moves stochastically according to a fixed policy away from the agent. The opponent moves away from the agent $80\,\%$ of the time and stays in the same cell otherwise. Our implementation of the opponent's movement policy varies slightly from the original paper allowing more movement away from the agent, thus making the scenario slightly more challenging. Section B.1 provides more details of the differences.

**RockSample.** The RockSample environment consists of a robot that must explore an environment and sample rocks of scientific value [38]. Each rock can either be good or bad and the robot receives rewards accordingly. The robot also receives a reward for departing the environment by entering an exit region. The robot knows the positions of every rock and its own location exactly, but does not know whether each rock is good or bad. The robot has a noisy sensor to check if a rock is good or bad and the accuracy of the sensor depends on the distance to the rock. Upon each use of the sensor, the robot receives a negative reward. In the following sections, a RockSample problem will be designated as RockSample$(n, k, sr, sp)$ where $n$ designates a grid size of $n \times n$, $k$ is the number of rocks, $sr$ is the sensor range efficiency, and $sp$ is the penalty for using the sensor. The reward for sampling a good rock and exiting the environment is $+10$ and the penalty for sampling a bad rock is $-10$.

## 4.2 Simulation Details

The simulation environment was built using the POMDPs.jl framework [39]. The Tag environment used the standard parameters and was simulated until the first of the agent tagging the opponent or 100 steps. The RockSample environment was simulated until the agent exited the environment. The SARSOP algorithm was used to generate policies for the agent. All of the agents used the same policies for the environment but incorporated the action suggestions differently. If the suggested action was the same as the selected action with the current belief, no modification of the belief was performed. This implementation decision potentially prohibits valuable information to be passed when actions align. However, simulations with the RockSample and Tag environments showed differences were negligible and the slight decrease in computation time allowed for more simulations to be performed. Performing belief updates with aligned suggestions would likely be critical in different scenarios.

In RockSample, the rocks were randomly initialized based on a uniform distribution. The agent's belief was initialized with a uniform distribution over the state space in Tag and a uniform distribution over the rocks in RockSample. The number of simulations for a given scenario varied and we provide the $95\,\%$ confidence interval for each value reported in the results section. The rewards reported are the mean value over all simulations for a given scenario. The number of suggestions refers to the number of times the suggested action differed from the action initially selected by the agent before considering the suggestion. To simulate a suggester that was not always present or could not communicate actions reliably, the simulation also supported varying the percentage of suggestions passed to the agent.

One suggester was used but the quality and consistency of the action suggestions varied. Different agents were simulated to evaluate our proposed approach and establish baselines in each scenario. Details of each agent are provided below.

- *Suggester.* The suggester is a combination of an all-knowing agent and a purely random suggester. The rate of randomness is adjusted to scale from purely random to completely all-knowing. If the suggester is sending a non-random suggestion, the action is selected from the POMDP policy using the true state of the environment.

- *Normal Agent.* The normal agent executes in the environment using the policy without considering any action suggestions. This agent provides a baseline when action suggestions are not considered.

- *Perfect Agent.* The perfect agent is initialized and executed with perfect knowledge of the state. At each time step an action is selected from the policy given the true state of the system. This agent provides and upper bound on the total expected reward executing with the given policy.

- *Random Agent.* The random agent chooses an action from a uniform distribution over the action space at each time step. This agent provides a lower bound on the total expected reward when investigating robustness to suboptimal and random action suggestions.

- *Naive Agent.* The naive agent executes in the environment using the POMDP policy. When action suggestions are received, it follows the action suggestions naively, and performs no modifications to its belief state. The rate at which the naive agent follows the suggestions was adjusted to investigate robustness. The rate of following the suggestion is depicted by $\nu$ in the results.

Table 1: Simulation results of various agents using an all-knowing suggester.

| Agent Type | Tag | | RS(7, 8, 20, 0) | | RS(8, 4, 10, −1) | |
|---|---|---|---|---|---|---|
| | Reward | # Sugg | Reward | # Sugg | Reward | # Sugg |
| Normal | $-10.7 \pm 0.3$ | – | $21.5 \pm 0.6$ | – | $10.1 \pm 0.1$ | – |
| Perfect | $-1.7 \pm 0.2$ | – | $28.4 \pm 0.5$ | – | $16.7 \pm 0.1$ | – |
| Naive | | | | | | |
| $\nu = 1.00$ | $-1.6 \pm 0.2$ | $3.7 \pm 0.1$ | $28.5 \pm 0.6$ | $15.3 \pm 0.3$ | $16.9 \pm 0.1$ | $8.4 \pm 0.1$ |
| $\nu = 0.75$ | $-3.8 \pm 0.2$ | $6.1 \pm 0.3$ | $26.0 \pm 0.2$ | $15.3 \pm 0.2$ | $14.6 \pm 0.1$ | $7.7 \pm 0.1$ |
| $\nu = 0.50$ | $-6.8 \pm 0.3$ | $15.2 \pm 0.9$ | $23.8 \pm 0.3$ | $15.1 \pm 0.2$ | $12.7 \pm 0.2$ | $7.8 \pm 0.2$ |
| Scaled | | | | | | |
| $\tau = 0.99$ | $-1.8 \pm 0.2$ | $3.1 \pm 0.1$ | $27.4 \pm 0.5$ | $6.4 \pm 0.1$ | $16.4 \pm 0.1$ | $2.8 \pm 0.1$ |
| $\tau = 0.75$ | $-2.4 \pm 0.2$ | $3.3 \pm 0.1$ | $27.3 \pm 0.5$ | $6.8 \pm 0.1$ | $16.2 \pm 0.1$ | $2.8 \pm 0.1$ |
| $\tau = 0.50$ | $-3.6 \pm 0.2$ | $3.9 \pm 0.1$ | $27.0 \pm 0.4$ | $7.8 \pm 0.1$ | $16.3 \pm 0.1$ | $3.2 \pm 0.1$ |
| Noisy | | | | | | |
| $\lambda = 5.0$ | $-1.8 \pm 0.2$ | $3.2 \pm 0.1$ | $27.5 \pm 0.4$ | $7.8 \pm 0.1$ | $16.4 \pm 0.1$ | $4.6 \pm 0.1$ |
| $\lambda = 2.0$ | $-2.0 \pm 0.2$ | $3.3 \pm 0.1$ | $27.8 \pm 0.6$ | $9.1 \pm 0.2$ | $16.3 \pm 0.1$ | $4.5 \pm 0.1$ |
| $\lambda = 1.0$ | $-2.4 \pm 0.2$ | $3.6 \pm 0.1$ | $26.8 \pm 0.6$ | $10.6 \pm 0.2$ | $16.2 \pm 0.2$ | $5.2 \pm 0.1$ |

- *Scaled Agent.* The scaled agent incorporates the methodology outlined in section 3.4 and uses eq. (3) to estimate $p(o_t^s \mid s_t)$. The hyperparameter $\tau$ is kept constant for each simulation and the value used is shown with the presented results. The value for $\tau$ was not adjusted to fine-tune performance. It was broadly changed to show overall effects of the hyperparameter in different situations.

- *Noisy Agent.* The noisy agent also incorporates the ideas from section 3.4. This agent uses eq. (4) to estimate $p(o_t^s \mid s_t)$. The hyperparamter $\lambda$ is kept constant for each simulation and the value is shown with the respective results. Like the scaled agent, the parameter $\lambda$ was not fine-tuned for performance. Coarse adjustments were made to depict how the hyperparameter influenced the algorithm.

## 4.3 Effectiveness and Efficiency Results

The different agents were compared using an all-knowing suggester with a $100\,\%$ message reception rate. Simulations were ran on Tag, RockSample$(7, 8, 20, 0)$, and RockSample$(8, 4, 10, -1)$. RockSample$(8, 4, 10, -1)$ was designed with the rocks near the corners of the environment. This layout emphasized the importance of each movement direction. The results are summarized in table 1. We also compared the different agents using suggesters with partial views of the state and provide those results in section A.

As expected, incorporating actions suggestions from an all-knowing suggester improves performance across all scenarios. Despite not directly following the suggested action, the Scaled and Noisy agents were able to reach a near perfect reward. A key metric to note with these results is the number of suggestions. The naive agent was able to achieve perfect scores by simply following the suggestions; however, it did not integrate any information contained from each suggestion and resulted in requiring more suggestions to achieve similar scores. While the scores for all assisted agents decrease when the hyperparameters change (become less trusting of the suggester) the naive agent's score decreases at a faster rate. Again, this difference is an artifact of the benefit of updating the belief of the agent with each suggestion.

A visual depiction of the change in the belief state of an agent in the Tag environment after incorporating an action suggestion is depicted in fig. 1. Figure 1a shows the belief the agent has of where the target is located. With that belief, the best action according to the agent's policy would be to move north. However, the agent receives a suggestion from a suggester to move west. Figure 1b shows the updated belief after using the noisy rational approach with $\lambda = 1$. There are multiple states to the west of the agent where the target might be located. The update process was able to incorporate

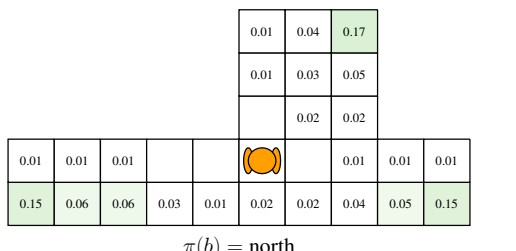

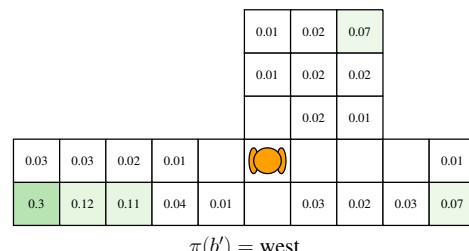



(a) Before received action suggestion.  (b) After belief update with $o^s = $ west.



Figure 1: Changes in the belief state of an agent after incorporating a recommended action. Using the original belief $b$ the agent's policy $\pi$ returns an action of *north*. The recommended action was to move west. Figure 1b depicts the updated belief $b'$ after incorporating the suggested action. After the update, the policy produces an action of *west*. The belief update used a noisy rational approach with $\lambda = 1$.

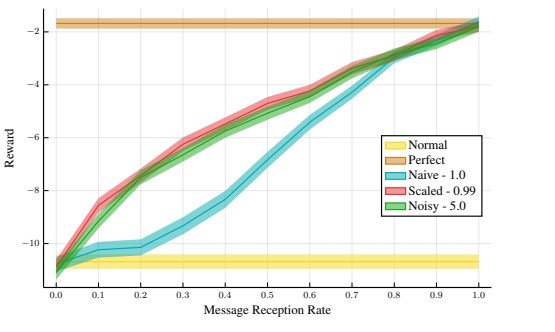

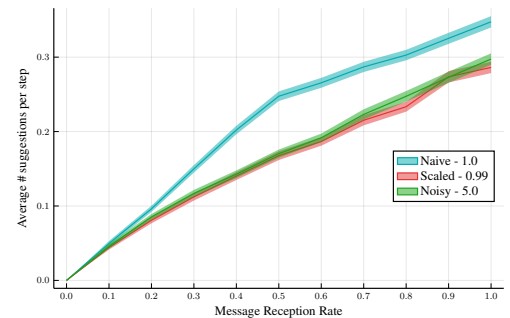



(a) Message rate versus mean reward.  (b) Message rate versus suggestions per step.



Figure 2: Tag performance with varying message reception rates.

the implied information of the *west* suggestion by shifting the distribution towards the west side of the grid.

Collaborators often are not capable of continuously providing suggestions to an agent. The percentage of suggestions received was varied to investigate the effectiveness of the agents in the presence of intermittent action suggestions. A suggester with perfect state knowledge was used but the message reception rate was varied. The results for the Tag environment are shown in fig. 2. The change in reward is shown in fig. 2a. Figure 2b shows how the ratio of the number of suggestions to the number of steps required to tag changes. The increased reward and lower suggestion ratio further emphasizes the benefit of our approach.

## 4.4 Robustness Results

The previous results assumed a perfectly rational suggester, but this is not always realistic. We evaluated the robustness of the different agents by adjusting the randomness of the suggester. The chance of a random action suggestion (uniformly picked from the action space) was varied from $0$ to $1$ representing a perfect suggester to a completely random suggester. The results from these simulations are shown for both the Tag environment and RockSample$(8, 4, 10, -1)$ in fig. 3.

As expected, the naive agent that follows all suggestions performs poorly as the randomness of the suggester increases. Decreasing $\nu$ increases the robustness, but sacrifices performance. The scaled and noisy approaches perform well even with high trust parameter settings ($\tau$ and $\nu$). These results demonstrate the value of not naively following suggestions and the benefit of balancing the agent's initial belief state with the information received from the action suggestion.

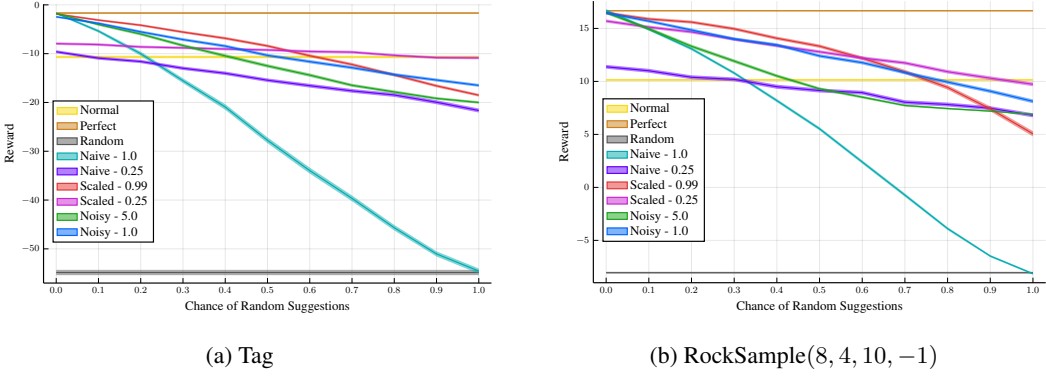

|  |  |
|---|---|
| (a) Tag | (b) RockSample$(8, 4, 10, -1)$ |

Figure 3: Robustness to suboptimal action suggestions.

# 5 Conclusion

We developed a new method to increase collaboration between heterogeneous agents through the use of action suggestions. Based on the idea that a suggested action conveys information about the suggester's belief, we used the agent's current policy to transform the suggested action into a distribution that we used to update the agent's belief. We applied this approach in simulation and demonstrated the increased efficiency by requiring fewer suggestions and suggestion rate. We further demonstrated our approach was robust to suboptimal decisions and could still perform better than an unassisted agent with more than $50\%$ of the suggestions being random. This methodology does not rely on knowledge of other agents. The only requirement is a shared objective and communication through the agent's set of actions. This low threshold of coordination increases collaboration opportunities with other systems including humans.

Integrating action suggestions as observations requires a change to the belief update process of the agent. The independence of the observations allow the update process to occur simultaneously or in any order which allows for asynchronous and unreliable suggestions without added complexity. This approach also scales linearly with the number of suggesters because the agent must only approximate the distribution over action suggestions for each suggester.

A critical assumption that enables this process with no knowledge of the suggester is that the suggester is collaborative. The scaled and noisy rational approaches do factor in suboptimal decisions but ultimately assume the suggester is providing actions in the best interest of the agent. If a suggester was maleficent, the belief would be skewed towards distributions that would explain the suggestions. The proposed approach is designed to be robust to suboptimal suggestions, but not resilient to bad actors.

There are many opportunities of future work expanding on these ideas. As previously discussed, this approach was formalized and applied to offline methods, but the same concept can be extended to online solvers. Online approaches also offer unique opportunities to use a suggested action to help balance exploration and exploitation while searching for an action. The experiment results demonstrated a trade-off of performance and robustness when changing the hyperparameters. Finding a way to learn the quality of the suggester and adjust the parameters real time is a promising idea. The benefit of modifying the agent's belief through an action suggestion is not limited to situations when the subsequent, updated belief is more accurate. A modified belief could be beneficial if the resulting actions are optimal regardless of the accuracy of the belief. This idea opens up possibilities of using action suggestions to increase performance when an agent's policy does not match the environment.

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
