# Appendix A   Different Quality Suggester Results

This section presents results on RockSample$(8, 4, 10, -1)$ when the suggester is not always all-knowing. In our approach, we formulated the belief update based on assuming the suggester observed the environment. These results demonstrate that our approach extends beyond an all-knowing suggester and can incorporate information from suggestions developed from different beliefs of the state.

Table 3 contains the mean rewards and table 4 contains the mean number of suggestions considered by the agent. The details of the agents are provided in section 4.2. In these simulations, instead of having an all-knowing suggester, the suggester maintained a belief on the state of the rocks and provided suggestions based on that belief. The suggester's initial belief varied and was determined based on two parameters. We represent a suggester with different parameters as (G | B). The parameters can be thought of as G being the initial belief that good rocks are good and B being the initial belief that bad rocks are good. For example, a suggester with parameters of $(1.0 \mid 0.0)$ would represent perfect knowledge of the environment while $(0.5 \mid 0.5)$ would be an initial belief of a uniform distribution. Examples of different initial beliefs based on different parameters are provided in table 2. In table 2, the examples are based on a true state of $[1, 1, 0, 0]$ where one represents a *good* rock and zero represents a *bad* rock.

Table 2: Example initial beliefs for different suggesters. The true state for this example is $[1, 1, 0, 0]$. Beliefs for $(0.75 \mid 0.25)$ are rounded to ease display.

| State | Suggester Initial Belief Parameters | | | |
|---|---|---|---|---|
| | $(1.0 \mid 0.0)$ | $(1.0 \mid 0.5)$ | $(0.75 \mid 0.25)$ | $(0.5 \mid 0.5)$ |
| $[0, 0, 0, 0]$ | 0.00 | 0.00 | 0.0352 | 0.0625 |
| $[0, 0, 0, 1]$ | 0.00 | 0.00 | 0.1055 | 0.0625 |
| $[0, 0, 1, 0]$ | 0.00 | 0.00 | 0.1055 | 0.0625 |
| $[0, 0, 1, 1]$ | 0.00 | 0.00 | 0.3164 | 0.0625 |
| $[0, 1, 0, 0]$ | 0.00 | 0.00 | 0.0117 | 0.0625 |
| $[0, 1, 0, 1]$ | 0.00 | 0.00 | 0.0351 | 0.0625 |
| $[0, 1, 1, 0]$ | 0.00 | 0.00 | 0.0351 | 0.0625 |
| $[0, 1, 1, 1]$ | 0.00 | 0.00 | 0.1055 | 0.0625 |
| $[1, 0, 0, 0]$ | 0.00 | 0.00 | 0.0117 | 0.0625 |
| $[1, 0, 0, 1]$ | 0.00 | 0.00 | 0.0352 | 0.0625 |
| $[1, 0, 1, 0]$ | 0.00 | 0.00 | 0.0352 | 0.0625 |
| $[1, 0, 1, 1]$ | 0.00 | 0.00 | 0.1055 | 0.0625 |
| $[1, 1, 0, 0]$ | 1.00 | 0.25 | 0.0039 | 0.0625 |
| $[1, 1, 0, 1]$ | 0.00 | 0.25 | 0.0117 | 0.0625 |
| $[1, 1, 1, 0]$ | 0.00 | 0.25 | 0.0117 | 0.0625 |
| $[1, 1, 1, 1]$ | 0.00 | 0.25 | 0.0352 | 0.0625 |

Table 3: Mean reward for various suggester agents on RockSample$(8, 4, 10, -1)$.

| Agent Type | Suggester Initial Belief Parameters | | | | | | | | |
|---|---|---|---|---|---|---|---|---|---|
| | $(1.0 \mid 0.0)$ | $(1.0 \mid 0.25)$ | $(1.0 \mid 0.5)$ | $(0.75 \mid 0.0)$ | $(0.75 \mid 0.25)$ | $(0.75 \mid 0.5)$ | $(0.5 \mid 0.0)$ | $(0.5 \mid 0.25)$ | $(0.5 \mid 0.5)$ |
| **Naive** | | | | | | | | | |
| $\nu = 1.00$ | $16.8 \pm 0.1$ | $16.5 \pm 0.1$ | $16.0 \pm 0.1$ | $13.7 \pm 0.1$ | $13.7 \pm 0.1$ | $13.0 \pm 0.1$ | $13.3 \pm 0.1$ | $12.9 \pm 0.1$ | $10.1 \pm 0.1$ |
| $\nu = 0.75$ | $14.6 \pm 0.1$ | $14.5 \pm 0.1$ | $14.0 \pm 0.1$ | $13.1 \pm 0.1$ | $13.0 \pm 0.1$ | $12.4 \pm 0.1$ | $12.7 \pm 0.1$ | $12.4 \pm 0.1$ | $10.3 \pm 0.1$ |
| $\nu = 0.50$ | $12.9 \pm 0.1$ | $12.9 \pm 0.1$ | $12.5 \pm 0.1$ | $12.2 \pm 0.1$ | $12.3 \pm 0.1$ | $11.9 \pm 0.1$ | $11.9 \pm 0.1$ | $11.8 \pm 0.1$ | $10.1 \pm 0.1$ |
| **Scaled** | | | | | | | | | |
| $\tau = 0.99$ | $16.5 \pm 0.1$ | $16.3 \pm 0.1$ | $15.2 \pm 0.2$ | $14.7 \pm 0.1$ | $14.6 \pm 0.1$ | $14.7 \pm 0.1$ | $14.1 \pm 0.1$ | $12.7 \pm 0.1$ | $10.1 \pm 0.1$ |
| $\tau = 0.75$ | $16.3 \pm 0.1$ | $16.2 \pm 0.1$ | $15.1 \pm 0.2$ | $14.7 \pm 0.1$ | $14.5 \pm 0.1$ | $12.8 \pm 0.1$ | $13.9 \pm 0.1$ | $12.7 \pm 0.1$ | $10.1 \pm 0.1$ |
| $\tau = 0.50$ | $16.3 \pm 0.1$ | $16.2 \pm 0.1$ | $15.3 \pm 0.2$ | $14.3 \pm 0.1$ | $14.4 \pm 0.1$ | $12.9 \pm 0.1$ | $13.7 \pm 0.1$ | $12.4 \pm 0.1$ | $10.1 \pm 0.1$ |
| **Noisy** | | | | | | | | | |
| $\lambda = 5.0$ | $16.4 \pm 0.1$ | $16.2 \pm 0.1$ | $15.3 \pm 0.1$ | $13.7 \pm 0.1$ | $13.4 \pm 0.1$ | $12.4 \pm 0.1$ | $12.8 \pm 0.1$ | $12.7 \pm 0.1$ | $10.2 \pm 0.1$ |
| $\lambda = 2.0$ | $16.4 \pm 0.1$ | $15.9 \pm 0.1$ | $15.1 \pm 0.1$ | $14.1 \pm 0.1$ | $13.4 \pm 0.1$ | $12.7 \pm 0.1$ | $13.6 \pm 0.1$ | $12.9 \pm 0.1$ | $10.1 \pm 0.1$ |
| $\lambda = 1.0$ | $16.4 \pm 0.1$ | $16.2 \pm 0.1$ | $15.4 \pm 0.1$ | $13.8 \pm 0.1$ | $13.7 \pm 0.1$ | $12.8 \pm 0.1$ | $13.4 \pm 0.1$ | $13.2 \pm 0.1$ | $10.1 \pm 0.1$ |

Table 4: Mean number of differing suggestions for various suggesters on RockSample$(8, 4, 10, -1)$.

| Agent Type | Suggester Initial Belief Parameters | | | | | | | | |
|---|---|---|---|---|---|---|---|---|---|
| | (1.0 \| 0.0) | (1.0 \| 0.25) | (1.0 \| 0.5) | (0.75 \| 0.0) | (0.75 \| 0.25) | (0.75 \| 0.5) | (0.5 \| 0.0) | (0.5 \| 0.25) | (0.5 \| 0.5) |
| **Naive** | | | | | | | | | |
| $\nu = 1.00$ | $8.40 \pm 0.10$ | $6.96 \pm 0.08$ | $6.39 \pm 0.09$ | $4.00 \pm 0.08$ | $2.97 \pm 0.04$ | $1.84 \pm 0.03$ | $3.20 \pm 0.08$ | $1.39 \pm 0.03$ | $0.0 \pm 0.0$ |
| $\nu = 0.75$ | $7.67 \pm 0.10$ | $6.36 \pm 0.09$ | $5.86 \pm 0.10$ | $3.99 \pm 0.08$ | $2.88 \pm 0.04$ | $1.93 \pm 0.04$ | $3.17 \pm 0.07$ | $1.59 \pm 0.04$ | $0.0 \pm 0.0$ |
| $\nu = 0.50$ | $7.58 \pm 0.15$ | $6.25 \pm 0.12$ | $5.54 \pm 0.12$ | $4.38 \pm 0.10$ | $2.94 \pm 0.05$ | $2.06 \pm 0.04$ | $3.36 \pm 0.09$ | $1.68 \pm 0.05$ | $0.0 \pm 0.0$ |
| **Scaled** | | | | | | | | | |
| $\tau = 0.99$ | $2.77 \pm 0.02$ | $2.69 \pm 0.02$ | $2.54 \pm 0.03$ | $3.55 \pm 0.04$ | $3.17 \pm 0.04$ | $2.97 \pm 0.04$ | $2.70 \pm 0.04$ | $2.20 \pm 0.03$ | $0.0 \pm 0.0$ |
| $\tau = 0.75$ | $2.78 \pm 0.02$ | $2.72 \pm 0.02$ | $2.57 \pm 0.03$ | $3.31 \pm 0.04$ | $3.02 \pm 0.04$ | $2.72 \pm 0.04$ | $2.65 \pm 0.04$ | $2.18 \pm 0.03$ | $0.0 \pm 0.0$ |
| $\tau = 0.50$ | $3.14 \pm 0.02$ | $2.95 \pm 0.03$ | $2.84 \pm 0.03$ | $2.97 \pm 0.04$ | $2.82 \pm 0.04$ | $2.50 \pm 0.05$ | $2.48 \pm 0.04$ | $1.97 \pm 0.03$ | $0.0 \pm 0.0$ |
| **Noisy** | | | | | | | | | |
| $\lambda = 5.0$ | $4.50 \pm 0.05$ | $4.12 \pm 0.05$ | $4.05 \pm 0.06$ | $3.39 \pm 0.04$ | $3.27 \pm 0.05$ | $2.94 \pm 0.05$ | $2.79 \pm 0.04$ | $2.16 \pm 0.05$ | $0.0 \pm 0.0$ |
| $\lambda = 2.0$ | $4.49 \pm 0.04$ | $4.23 \pm 0.03$ | $3.85 \pm 0.04$ | $3.05 \pm 0.04$ | $2.52 \pm 0.03$ | $2.20 \pm 0.04$ | $3.04 \pm 0.04$ | $3.04 \pm 0.05$ | $0.0 \pm 0.0$ |
| $\lambda = 1.0$ | $5.22 \pm 0.04$ | $4.41 \pm 0.04$ | $3.98 \pm 0.05$ | $3.21 \pm 0.06$ | $2.31 \pm 0.04$ | $1.59 \pm 0.03$ | $2.77 \pm 0.05$ | $2.10 \pm 0.05$ | $0.0 \pm 0.0$ |

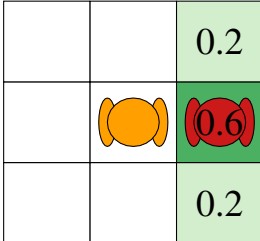

(a) Legacy Implementation.

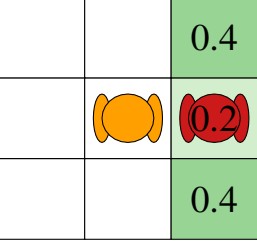

(b) Our Implementation.

Figure 4: Example demonstrating the differences in the transition probabilities for the opponent in Tag. The agent is depicted in orange and the opponent is shown in red. The numbers are the probability that the opponent moves to that grid square. Figure 4a shows the transition probabilities for the legacy implementation and fig. 4b shows the transition probabilities of our implementation.

## Appendix B   Environment Differences

### B.1   Tag Differences

Our implementation of the Tag environment differs slightly from the original implementation. The difference is in the transition function and results in our scores differing from those presented in other papers [30], [33]. The original problem described the opponent's transition as moving away from the agent $80\,\%$ of the time and staying in the same cell otherwise [30]. The original implementation considered all actions away from the agent as valid and did not consider if an action would result in hitting a wall. Additionally, the actions were considered in pairs (i.e. east/west and north/south) with $40\,\%$ allocated to each pair. This implementation choice often results in the opponent staying in the same cell greater than $20\,\%$ of the time despite having valid moves away from the agent.

Our implementation only considered actions that would result in a valid move away. The move away probability was distributed equally to each valid action. This change results in the opponent moving away from the agent more than the original implementation and results in a slightly more challenging scenario. An example of the differences in the opponent transition probabilities in a $3 \times 3$ environment is shown in fig. 4.

### B.2   RockSample Additions

Our implementation of the RockSample environment is the same as previous work except for an additional penalty for performing the *sense* action (i.e. we added a penalty when the agent uses the sensor to gather information about a rock). We added this penalty when formulating a problem to emphasize the importance of each action selection. Our implementation is identical to other works when the penalty is set to zero and can be verified by our RockSample$(7, 8, 20, 0)$ experiments.