# OpenReview forum: "Collaborative Decision Making Using Action Suggestions"
_NeurIPS.cc/2022/Conference — NeurIPS 2022 Accept_

### Official Review · Reviewer_pPmT · 2022-07-11

**Rating:** 6
**Confidence:** 3
**Soundness:** 3 good
**Presentation:** 4 excellent
**Contribution:** 3 good

**Summary:**

This paper presents a model and experiments to demonstrate the usefulness of (possibly intermittent and/or incorrect) human action suggestions on a POMDP-based system, treating the suggestion not as a direct input but rather as an indicator of the partially-observed implicit state. Simulated experiments on a discrete state problem with an offline generated policy show some improvement using (possibly intermittent and/or partially incorrect) action suggestions from a human with an omniscient view (when sending correct information).


**Questions:**

How do these experiments shed light on your original research impetus of reducing brittle behavior via human suggestions ?

Is the assumption that the “suggested action is not influenced by the previous action” really reasonable, given that one might think the human would be especially wanting to suggest an action if the agent does something bad/wrong/ill-advised?


**Limitations:**

This work seems theoretical enough to not have significant societal impacts (beyond, perhaps, the use of computational agents, but here they have human supervisors!)

**Strengths And Weaknesses:**

Originality
The authors point out much related work; the specific contribution here is that the action is not taken itself as a suggestion to be used or ignored, but rather as another kind of observation that can update the agent’s estimate of the true state, thus potentially wringing more information from the suggestion than merely following it would (and being more helpful when the suggestions are intermittent).

Clarity
The paper is generally clear and easy to read. The details of computation of the scaled rational and noisy rational approaches are a bit unclear, but the authors claim to provide code in supplemental material.

I strongly suggest that the authors mention early-on (e.g. in the abstract!) that the human is the one giving the suggestions (one could also view that multiple agents who have distinct partial views of the global state could also provide such suggestions that then fold into the local estimate of the full state, which is what I originally thought when I read the abstract. The paper *does* make this clear in the last line of paragraph 2 in the introduction.

Last paragraph section 2 “we do not build an explicit model” —> “we do not build an explicit model of the suggester”

I assume the notation “o_{o:t}” means all observations from time 0 through the current time?

Quality
The abstract and introduction talk a lot about failure and brittle or inconsistent behavior of current MDP-variant-based systems, and that human suggestions could help with that. However, the experimental environments (Tag and RockSample) don’t clearly show these issues. Instead, the versions of the system that take into account the *omniscient* collaborator, who has information not available to the agent, do better (even the naive agent that simply uses suggestions directly). The authors do go to pains to demonstrate that the more complex indirect use of the intermittent (but often omnisciently correct) suggestions as observation about true state require fewer communications and more robust in the face of unreliability of the suggester, but how that might help reduce brittle and inconsistent behavior as discussed in the abstract and introduction is somewhat unclear.

[After rebuttal: just downplay this, especially in abstract/intro, so the reader is not looking at this as the primary contribution. Could mention it later in discussion as a possible area to explore]

Aside from that: how many runs of each scenario? The text just says that they “varied” (why not some single, consistent numbers runs?)

Significance
The core idea seems to be a useful one, although the demonstration here is on small, discrete, 2-agent environments without catastrophic failures, brittle behavior, or clear inconsistencies.

---

> ### Author Response · Authors · 2022-07-30
> **Response to Reviewer pPmT**
>
> Thank you for the time and effort you put into the review, especially the considerations and questions on how our approach relates to our statements of brittle behavior of systems.
>
> ### Response to Question 1
>
> The domains picked were simple domains and, as you stated, do not directly show the issues related to failures and brittleness. However, these domains are classic domains that readers are familiar with and allow the focus to be on the performance of the method. We attempted to choose experiments that emphasize certain attributes of our approach where these attributes would support increased performance in domains where MDP/POMDP solutions experience more failures.
>
> We could have chosen a more complex scenario; however, a lot of design choices could change the performance of an agent without even considering suggested actions. We concluded it would be easier to compare differences and the benefit of the approach in simple domains where specifics of the POMDP/MDP solutions were not as relevant.
>
> As you stated, we do make an effort to show our method is robust to an unreliable suggester and intermittent suggestions. Section 4.3 shows the increased efficiency of our method compared to naively following the suggestions. This section provides evidence that the extra effort and computation is worth doing even when a simple method like a naive approach is available. Section 4.4 shows that the approach is robust to intermittent and suboptimal suggestions. This section suggests that even in cases where you might have only a few suggestions or suboptimal ones, the system benefits from these extra ''observations.''
>
> From the results, we concluded that integrating action suggestions help a system and in situations where the system is brittle or failing, this would be a straightforward way to provide ''corrections'' to a system without taking over control. While we did not directly show our approach on more challenging domains, we believe the set of experiments allows the reader to reach similar conclusions and focus on the merits of the approach while removing domain-specific variables that might influence performance. If you still think the connection is unclear, we can expand upon the connection of the results to our original motivation.
>
>
> ### Response to Question 2
>
> There are situations that seem practical where the suggestion would be based on previous actions. We considered situations where this might be the case but decided a Markovian approach would be more robust and still allow for a suggester to intervene.
>
> Assuming the suggested action is not influenced by the previous action and only on the current state still allows for the case where the robot is doing something bad/wrong. We believe it is reasonable to assume the suggester would send the best action based on its own policy considering only the state. From this perspective, we are assuming even if the suggester decided to provide a suggestion based on previous actions, the current suggestion would still be the ''best'' action from the suggester.
>
> We do not consider the timing of when the agent receives a suggestion in our formulation. Considering the timing of received suggestions would allow for more influence of the suggester, but would likely sacrifice robustness. We believe this is an interesting area to research and consider when the suggester is reliable and robustness is not much of a concern.
>
>
> ### Responses to other Questions
>
> **How many runs of each scenario?**
>
> We performed the majority of our runs with a set number. We varied the number in some runs based on computational burden and time. Although we varied the run numbers, we reported a 95% confidence interval for all results.
>
> **Notation “o_{o:t}”**
>
> Yes, the subscript notation 0:t refers to all instances of that variable from time = 0 to time = t.
>
> ### Other Comments
>
> We do focus on using human suggestions, especially as motivation, as we think our approach allows for a straightforward way of integrating human decision-making. However, we try to stay abstract and emphasize that the suggester can be anything, just that it needs to be collaborative and communicate through valid actions. We focus on the case of a single agent providing suggestions, but expanding this concept to more suggesters is straightforward and linear in complexity. We have considered the case of multiple collaborative agents that have partial views of the global state and currently have work in progress that expands on this concept more.
>
> While we do use the assumption that the action suggestion came from a suggester that observed the state directly when we update the belief, we do not limit the method to only all-knowing agents. See the response to question 3 from reviewer Fv1x for more on this topic.
>
> ### Conclusion
>
> Again, thank you for the review and the questions. We believe we have addressed all of your concerns and questions. Please let us know if there is anything we can clarify or expand upon.

---

> > ### Comment · Reviewer_pPmT · 2022-08-08
> > **Response to Response**
> >
> > Thanks for your detailed comments. All your responses make sense (although you are often saying that the system would also work under these different, less general, circumstances without demonstration), that's fine, and I also suspect you are correct.
> >
> > I think my overall issue was misjudging the overall intent of the paper by the abstract and introduction as focussed on brittleness, inconsistency, and failures. Instead, I'd say the paper is focussed on using (collaborative) action suggestions to increase base performance (not specifically focussed on avoiding failures by some definition, or avoiding brittleness to some external change), even if those collaborative suggestions are intermittent and unreliable (the suggestions are also not tied to specific failures or undesirable learned behaviors). With that in mind, I think the approach looks quite interesting (especially with respect to multi-agent activity with partial observability, a classic problem).

---

> > > ### Author Response · Authors · 2022-08-09
> > > **RE: Reply**
> > >
> > > Thank you for the time and effort of the review and the response!
> > >
> > > Based on the provided feedback, we have uploaded our updated paper with supplemental material and other minor changes. We have included experimental results of a suggester with partial beliefs of the state in the supplemental material found after the references and checklist.
> > >
> > > Thank you again for the review and feedback.

---

> ### Author Response · Authors · 2022-08-07
> **Author Follow Up**
>
> As the discussion period is coming to an end, we wanted to thank you again for the review and suggestions. If there are any questions remaining or areas we can help clarify, please let us know.

---

### Official Review · Reviewer_Fv1x · 2022-07-12

**Rating:** 7
**Confidence:** 4
**Soundness:** 3 good
**Presentation:** 3 good
**Contribution:** 3 good

**Summary:**

The paper looks at the problem of designing a framework to support collaboration between an autonomous system and a human supervisor. In particular, the human supervisor provides action suggestions to the autonomous agent to help overcome any limitations the human could have in its sensor capabilities. The human is assumed to be a potentially suboptimal agent with full observability, while the agent is expected to operate under partial observability. The method proposes to use the action suggestions, by using the human policy as part of the observation model of the underlying POMDP, where the actions suggested by the human would correspond to the full observation. This means that the human suggestions could be used to refine the belief space the agent may hold regarding the current state. However, when implementing the approach, rather than directly using this new observation model during the POMDP planning, they use an approximation where in the human suggestion is used to further refine the belief space only if the suggestion differs from the current policy action. This approximate method is then tested on two benchmark domains (under various conditions), where the paper demonstrates that the human suggestions does make a positive impact on the system performance.

**Questions:**

I would appreciate it if the authors could respond to the weaknesses and the clarification I had mentioned in the previous section. To repeat these are,
1. The merit in the assumption that the human would be a supervisor who constantly observes the system as opposed to other interaction paradigms
2. How the choice to only use the human suggestion to update belief states when they differ from the robot’s policy may have affected the results in section 4.4?
3. Please clarify, what you mean by the statement that the method doesn’t build a model of the human


**Limitations:**

I don’t see any specific ethical concerns with the proposed methods. I have listed some of the limitations I have noticed of the proposed method under the weakness and other suggestions section. Most of these relate to practical challenges of interacting with humans. I would strongly urge the authors to consider checking human-factors literature on creating systems that are meant to effectively interact with people.

**Strengths And Weaknesses:**

Strengths:
- Developing methods that are able to effectively collaborate with people and incorporate their suggestions into their decision-making process is an important problem.
- The paper presents a relatively clean and clear formulation. Additionally, the fact that the formulation allows the problem to be directly compiled into a standard POMDP means that one could always leverage the latest developments in POMDP planning/RL for the problem.
- While the evaluations are limited they do present some intriguing results. Specifically, the results presented in section 4.4 in regards to robustness of the methods to suboptimal human behavior is interesting.

Weaknesses:
I have two big concerns of the paper:

- Type of Interaction: To me the assumption that a human would be an ever present supervisor ready to provide suggestions to the system seems like the worst way to make use of the human. There are many results from psychology and human-factors (not to mention many high profile accidents) that show that humans are pretty bad in roles where they are expected to constantly pay attention and intervene as required. Humans constantly fall prey to issues like automation bias, where they may overtrust the system when they see that the system is performing as they expected in the beginning. I would argue that it would be better to treat the human as an expensive but highly information source of knowledge about the task. Rather than asking the human to come up with an action, the system could have equally asked the human about the underlying state. To avoid the system bothering the human too much, one could also attach some high cost to such information gathering actions.

- Use of the approximation: It's quite unfortunate that the authors chose not to use the exact formulation for the system evaluation. As the paper itself points out, the human agreeing with the system could also present a lot of useful information. While it is a positive sign that the system is still performing well even while only using a subset of the information, it is not quite obvious to me how the inclusion of this additional information would have impacted the results in section 4.4.


Clarification:
- Claims about human model: In multiple places in the paper, the authors claimed that unlike earlier works the proposed method builds no explicit human model (repeated in both section 2 and 3.3). As far as I can tell this is not true. It is true that the proposed method doesn’t make strong assumptions about the policy and in some sense the rationality of the human, however, you do make strong assumptions about the sensor model and the knowledge the human supervisor possesses. The authors assume that the human’s knowledge about the task dynamics is the same as what the system’s dynamics is, or at least that the system knows this model. So in some sense you don’t have to build a human model, because it's known beforehand. Additionally, to generate the human policy from this model you make additional assumptions about the rationality of the human planner. It’s also worth noting that there are many previous works (like [a] and [b]) that don't assume that the human’s knowledge about the task dynamics is the same as what the system knows.



Other Suggestions
- User Study: While the current experiments are a good starting point, I would strongly urge the authors to perform user studies to see how effectively their system can actually work with real people.
- Other forms of information: There is no reason to believe that the information about the underlying state is the only kind of information the human can provide. In future versions, I would suggest the authors consider cases where the human feedback may also reveal additional information about the task, including information about the task reward or potential information about effective strategies the system could use to solve the task.

[a] Reddy, Sid, Anca Dragan, and Sergey Levine. "Where do you think you're going?: Inferring beliefs about dynamics from behavior." Advances in Neural Information Processing Systems 31 (2018).

[b] Gong, Ze, and Yu Zhang. "What is it you really want of me? generalized reward learning with biased beliefs about domain dynamics." Proceedings of the AAAI Conference on Artificial Intelligence. Vol. 34. No. 03. 2020.

---

> ### Author Response · Authors · 2022-07-30
> **Response to Reviewer Fv1x**
>
> Thank you for the time and effort you put into the review, especially the considerations on interacting with humans.
>
> ### Response to Question 1
>
> A human providing constant supervision was not our thought or intention with this paper or approach. In fact, the idea that the human is a valuable and finite resource was a large motivation to investigate how to more efficiently use each human input.
>
> We state in the abstract ''A way to mitigate the risk of failures is to integrate human oversight of the autonomous systems and rely on the human to take control when the autonomy fails.'' That statement was meant to provide a current method that is employed to mitigate failures of autonomous systems. Our intent was for the current method (of constant human oversight) to motivate our approach - which does not require constant oversight to achieve similar performance as continuous monitoring and naively following directions.
>
> We can see how that statement and the flow of the abstract can be misleading. We will work to reword the abstract to make it more clear that we are proposing a method that enables intermittent oversight and more efficiently integrates each input.
>
> ### Response to Question 2
>
> We did investigate incorporating the suggestions even if they agreed with the robot's policy. In all the scenarios we investigated, the performance slightly improved or was within the confidence interval.
>
> We anticipated similar performance and more robustness to the random suggester experiment but did not run the simulations initially. When the suggestions aligned with the agent, we would expect the belief to be ''strengthened'' and thus more robust to subsequent differing suggestions. We ran those experiments again but allowed the agent to incorporate every suggestion. The table below shows results for a 50% random suggester. These results support our claim that performing belief updates with aligned suggestions would be beneficial in different scenarios.
>
> #### **Chance of Random = 50%**
> | Agent | Mean Reward, All | Mean Reward, Differ |
> | -- | -- | -- |
> | Scaled 0.99 | 14.22 +/- 0.12 | 13.33 +/- 0.16 |
> | Scaled 0.25 | 13.54 +/- 0.12 | 12.79 +/- 0.15 |
> | Noisy 5.0 | 10.04 +/- 0.10 | 9.30 +/- 0.14 |
> | Noisy 1.0 | 12.80 +/- 0.13 | 12.40 +/- 0.16 |
>
> ### Response to Question 3
>
> When we state we do not build an explicit model of the suggester agent, we mean that we only use the knowledge of the autonomous agent's policy and environment along with the assumption that the suggester is collaborative. We were trying to convey the idea that we do not need any information from or about the suggester agent and use only information from the autonomous agent. We debated internally on the appropriate way to deliver this message and will look into our wording again.
>
> When an agent receives a suggested action, it uses its own policy and environment representation to reason over the suggestion. By doing this, we are representing the suggester implicitly. As you pointed out, this rationale involves assuming the suggester has the same policy as the agent and makes suggested actions based on observing the state.
>
> Despite making the aforementioned assumptions to implicitly model the suggester, it still performs well when those assumptions are not valid. We performed experiments where the suggester was also in a partially observable environment but had information different from the agent (e.g. the suggester had a higher initial belief about good rocks). Instead of approaching a *perfect* score, the results approached the score based on the weighted initial belief at a similar rate. We also performed experiments where multiple suggesters with varying information provided action suggestions and where the suggester had a policy of an environment different from the agent. In each scenario, the results were improved with suggestions. We did not include those experiments based on page limitations and the desire to investigate those situations more. We can provide results from partially observable suggesters in the appendix if you think that would add value.
>
> ### Other Comments
> Thank you for the suggestions for a user study and for incorporating other information beyond the state. I believe we are aligned in viewing humans as a valuable, but limited resource. We are interested in expanding ways where we can take advantage of human/robot collaboration.
>
> Our approach is not limited to a human suggester agent and can extend to any agent that can provide recommended actions. We use the idea of human and robot collaboration to show the versatility of our approach. While we do use the assumption that the action suggestion came from a suggester that observed the state to update the belief, we do not limit the method to only all-knowing agents.
>
> ### Conclusion
>
> Again, thank you for the review and the questions. We believe we have addressed all of your concerns and questions. Please let us know if there is anything we can clarify or expand upon.

---

> > ### Comment · Reviewer_Fv1x · 2022-08-07
> > **Re: Response**
> >
> > Thank you so much for the detailed response and for running the additional experiments. Most of my primary concerns have been addressed by the authors responses. However, I still want to mention that while the human supervision isn't constant, as I understand it the system still places the onus on the human to intervene and provide information. This might not be ideal especially in mission-critical scenarios, where the human not-intervening or providing information could be risky. As such, I would still recommend methods where the system may actively query the user for information as required (say if the uncertainty is high or based on some formalization of risk). However, as the first step in the direction I think the method is completely acceptable and I will increase my score to an accept.

---

> > > ### Author Response · Authors · 2022-08-09
> > > **RE: Reply**
> > >
> > > We agree that there is a lot of potential in methods that can actively query the user for information and then integrate the recommendation in a smart way. We hope to expand on these ideas and improve safety by intelligently integrating humans into the process without placing the burden of intervention on them.
> > >
> > > Based on the provided feedback, we have uploaded our updated paper with supplemental material and other minor changes. We have included experimental results of a suggester with partial beliefs of the state in the supplemental material found after the references and checklist.
> > >
> > > Thank you again for the review and feedback.

---

> ### Author Response · Authors · 2022-08-07
> **Author Follow Up**
>
> As the discussion period is coming to an end, we wanted to thank you again for the review and suggestions. If there are any questions remaining or areas we can help clarify, please let us know.

---

### Official Review · Reviewer_pmf6 · 2022-07-14

**Rating:** 5
**Confidence:** 5
**Soundness:** 2 fair
**Presentation:** 3 good
**Contribution:** 2 fair

**Summary:**

This paper describes an approach for collaborative decision-making where another agent can make suggestions (in the form of suggested actions) to the subject agent in order to influence their chosen actions, but the subject agent retains autonomy in its own decision-making.  In particular, the subject agents decision-making is modeled as a POMDP and the suggestions from other agents are taken as observations by the subject agent and incorporated in the agent's belief state revision.  Two approaches are provided for representing the stochastic observation function for suggested actions, and empirical evaluation on the Tag and variants of the RockSample domains are employed, demonstrating how incorporating suggested actions is beneficial to the subject agent's utility returns.

**Questions:**

1) How does the agent calculate a Q value for a state instead of a belief (to be used in Noisy Rational)?  Could this approach generalize to algorithms other than point based value iteration derived solutions (e.g., SARSOP)?  Especially since the vast majority of POMDP research is using online planning using Monte Carlo methods, which is vastly different (and maintains no representation for Q values over states)?

2) From what I remember, a key feature of the SARSOP algorithm compared to previous PBVI-based algorithms is its focus on planning only for beliefs that are reachable from the initial belief.  This gave it a computational speedup in that it could prune out unreachable beliefs and not bother planning for them (in its point-based backups).  However, by adding new observations during runtime (i.e., suggestions from the other agent), you've changed the set of reachable beliefs -- a suggestion might cause the agent's new belief to not be near the ones considered during SARSOP.  In that case, your alpha vectors from SARSOP might not be a good approximation of the value function at your new, previously unreachable belief, and thus we have a new source of approximation error.  What impact might this have on the subject agent's performance in your approach, and how might it be prevented or mitigated?

3) Do you have an idea of why your SARSOP ("Normal") performance on Tag is so much lower than originally reported?

4) I appreciate that you included the code of your experiments, but some details of how the domains were changed from their typical implementations would be helpful in the paper.  For example, you state that you changed the opponent behavior in Tag to allow "more movement away from the agent", but what does that mean?

**Limitations:**

The limitations of the approach are adequately described.

**Strengths And Weaknesses:**

The problem of collaborative decision-making is relevant to the multiagent planning community at NeurIPS.  The paper is relatively easy to follow.  I appreciated that it included more than one experimental domain, although the problems are relatively and old (see the POMCP, DESPOT, and other papers for larger, more recent benchmark domains).  The paper was also thorough in testing different parameters to the settings in the experiments (e.g., the \nu, \tau, and \lambda parameters).  I would have appreciated also seeing some theoretical analysis of the approach, such as regret bounds when the two different approaches for estimating the observation function are used.

I also had some concerns with the approach and its performance.  The Noisy Rational estimate of p(o^s_t | s_t) requires the subject agent to have access to Q(s_t, o^s_t).  However, in POMDPs, the agent typically constructs Q(b, a) not Q(s, a).  So I'm not sure when or how the agent will have access to Q(s, a) to use the approach.  Also, in the original SARSOP paper, the average performance on Tag was -6.13, whereas you've reported -10.7, which calls into question whether (1) your implementation is correct, or (2) you ran it with a reasonable set of hyperparameters (e.g., number of backups).

### Post Rebuttal Period ###

I thank the authors for all of their responses to the reviewers questions and comments, including my own.   I think the idea of treating recommended actions as observations is a clever way to adapt single-agent POMDPs to incorporate advice from another agent (human or artificial), and I think the approaches you proposed for defining an observation function over those new observations are reasonable first approaches (and could inspire additional mechanisms with beneficial properties). The experiments and empirical results were also extensive and used reasonable benchmark problems and baselines. At the same time, without any theoretical analysis (which is increasingly common in papers like this), it is difficult to know how the results might generalize to other domains and situations.  I strongly encourage you to continue this work, and if you were to add some theoretical analysis, I think it would be a valuable contribution to the literature.

---

> ### Author Response · Authors · 2022-07-30
> **Response to Reviewer pmf6**
>
> Thank you for the time and effort you put into the review, especially the critical thought on our assumptions within the POMDP framework.
>
> ### Response to Question 1
>
> Using an alpha-vector policy, we can calculate $Q(s, a)$ by performing a one-step lookahead using a belief that the agent is in state $s$.
>
> We start with
> $$
> Q(b,a) = R(b,a) + \gamma \sum_o \sum_s \sum_{s^\prime} T(s^\prime \mid s, a) O(o \mid s^\prime, a) b(s) \mathcal{U}^{\Gamma}(b^\prime)
> $$
> where $b^\prime = \text{Update}(b, a, o)$. If we are looking for the value function from a particular state $s^*$, we can start from that state. That is, let $b(s^*) = 1$. Then we get
> $$
> Q(s^*,a) = R(s^*,a) + \gamma \sum_o \sum_{s^\prime} T(s^\prime \mid s^*, a) O(o \mid s^\prime, a) \mathcal{U}^{\Gamma}(b^\prime)
> $$
> where we can calculate $\mathcal{U}^{\Gamma}(b^\prime)$ by $\max_{\alpha \in \Gamma} \alpha^\intercal b^\prime$.
>
> This particular method cannot be directly applied to algorithms like online planning using Monte Carlo methods, but the concept can be extended. For example, in POMCP, particles are sampled from the history to estimate the belief. We could estimate the value function for a given state by keeping track of the values of samples from that state. While certain implementation details might result in more complexity, the concept of using the suggested action and the agent's current policy (e.g. a searched tree) would extend. We left the details of extending this concept to online approaches for future work.
>
> ### Response to Question 2
>
> SARSOP and similar algorithms do gain efficiency by performing backups only on reachable beliefs from an initial starting belief. However, SARSOP differs from other PBVI methods by focusing on the space reachable under an optimal policy (versus some arbitrary sequences of actions).
>
> The algorithms use the concept of sampling from a reachable set, but the sampling and pruning are still based on heuristics. For example, HSVI2 samples to minimize the gap between upper and lower bounds on the optimal value function and FSVI uses an MDP to guide sampling. SARSOP similarly uses heuristics but also considers when the computed policy would take it outside the optimal reachable belief space. So instead of pruning alpha vectors that were dominated, the authors introduced a margin of error and referred to it as $\delta$-dominance.
>
> We agree that adding observations will change the reachable set. However, based on the process of sampling and pruning we do not believe there is a significant effect on our approach. While there are some cases where this might be more of a factor, we could avoid this situation by using a solution method that considers a larger portion of the belief space. One approach would be to increase the $\delta$-dominance factor in SARSOP.
>
> ### Response to Question 3 and Question 4
>
> The difference in performance on Tag is due to the change in the transition function associated with the opponent. We will add the details of the differences in an appendix (including the additions we made to RockSample).
>
> The original paper for Tag states that the opponent moves away from the robot with Pr = 0.8 and stays in place with Pr = 0.2 (Pineau et al. 2003). The implementation used by the SARSOP paper can be found at https://bigbird.comp.nus.edu.sg/pmwiki/farm/appl/index.php?n=Main.Repository.
>
> The SARSOP implementation considered actions in the north/south and east/west directions separately. If an action resulted in hitting a wall, the probability was redistributed to the ''do not move'' action. This implementation resulted in the opponent often staying in place with a higher probability than the stated $0.2$.
>
> Consider a 3x3 grid where the (1, 1) position is the bottom left and the (3, 3) position is the top right. Let the agent be at (1, 3) and the opponent at (2, 3). Under the original implementation the opponent transition probabilities would be
> * Move to (3, 3): $0.4$
> * Move to (2, 2): $0.2$
> * Stay in place: $0.4$
>
> This is based on them considering both moves of north and south as valid moves even though moving north would result in hitting a wall and thus staying in the same location.
>
> Our implementation differed by finding all *valid* moves away from the agent and distributing the move away probability equally to each valid move. So in the same example as above, our implementation would have the opponent transition probabilities as
> * Move to (3, 3): $0.4$
> * Move to (2, 2): $0.4$
> * Stay in place: $0.2$
>
> We ran the original implementation and achieved similar results as the original SARSOP paper (-6.07470 +/- 0.16212). We provide a transition function that is the same as the original and a script to rerun Tag here: https://osf.io/bn739/?view_only=7c11c564e1084a1f98ba3bdb37ba2568.
>
>
> ### Conclusion
>
> Again, thank you for the review and the questions. We believe we have addressed all of your concerns and questions. Please let us know if there is anything we can clarify or expand upon.

---

> > ### Comment · Reviewer_pmf6 · 2022-08-08
> > **RE: Authors' Responses**
> >
> > I thank the authors for their responses to my questions and comments.  I think my questions have been addressed.
> >
> > My main remaining question is whether the authors can summarize the significance of the work for me.  Taking a high level look at the work, it appears that the authors demonstrate:
> >
> > 1) If the agent has access to a perfect oracle that suggests optimal actions, the agent improves its behavior by listening to the oracle.  Several approaches for incorporating information from this perfect oracle are considered, and all are beneficial.
> >
> > 2) If the oracle is no longer perfect, the agent's performance in listening to the oracle changes in ways that are proportional to the noise in the oracle's suggestions.
> >
> > Both of those are rather predictable results, albeit it is important to confirm them empirically, and the use of standard benchmarks is a good experiment design.  Having several approaches for incorporating information is helpful, but I didn't get a lot of insight into when one should be used over the other, and neither are particularly complicated in their design or implementation.  In practice, a perfect oracle won't be available (or planning wouldn't be needed), and I didn't get a strong sense of what should be done in real-world settings under those conditions.  Relatedly, the work also lacks theoretical analysis, which could be helpful in establishing performance guarantees of each approach to incorporating information (e.g., how robust can each be to noise in the oracle?).
> >
> > Is there anything I am missing that further highlights the significance or contributions of the work?  I think it is potentially useful, but I am also left with a lot of questions about what would happen if this were applied in real-world situations without a perfect oracle.

---

> > > ### Author Response · Authors · 2022-08-09
> > > **RE: Subsequent Response**
> > >
> > > Thank you for the follow-up questions and for allowing us the opportunity to respond.
> > >
> > > We agree that it is rather predictable that following optimal actions from a perfect oracle would improve performance. Additionally, if the oracle is no longer perfect, it does make sense to expect to see the performance degrade. In fact, the situation of just blindly following the suggested actions is the Naive agent (with $\nu=1.0$) in our work.
> > >
> > > Our primary contributions can be summarized in two main points.
> > > 1) We demonstrate a way that we can view the suggested actions as independent observations of the environment and modify the belief update process.
> > > 2) We provide two methods to estimate the distribution over suggested actions using the agent's policy, thus enabling a way to update the agent's belief without explicit knowledge of the suggester.
> > >
> > > By using the suggested action as an observation, the agent more efficiently uses each suggestion. While naively following a perfect oracle increases performance, our proposed methods require fewer suggestions (section 4.3) while achieving similar performance. Furthermore, by integrating the suggested action by modifying the belief of the agent, we also gain robustness to suboptimal suggestions. When integrating suggestions, the performance of the agent does decrease proportional to noise in the oracle's suggestion. However, our proposed methods achieve greater robustness with performance decreasing much slower (section 4.4).
> > >
> > > We formulate our approach using the assumption that the suggester provides an action based on the state of the environment. This allowed a simple update procedure without explicitly modeling the suggester (e.g. we do not have to model the observation capability of the suggester). The experiments we provide in the paper all use a variation of an all-knowing suggester. However, we do not limit the method to only all-knowing suggesters. This type of suggester followed closely with our assumptions and allowed us to emphasize the effectiveness and robustness of our approach.
> > >
> > > We performed experiments where the suggester was also in a partially observable environment but had information different from the agent (e.g. the suggester had a higher initial belief about good rocks). We viewed this type of suggester as more aligned with a real-world setting. Instead of approaching a perfect score, the results approached the score based on the weighted initial belief at a similar rate. Based on page limitations, we initially did not include those results. However, we have now added the results of different quality suggesters as supplemental material.
> > >
> > > The two methods, scaled rational and noisy rational, are related in concept but do make different assumptions about the rationality of the suggester. We agree that there is an opportunity to expand on this idea and provide theoretical analysis to help provide insight on when one might be better than the other. Additionally, we believe there would be value in a more theoretical analysis on the assumption the suggester makes suggestions based on observations of the state. However, we have left these research opportunities for future work.
> > >
> > > Based on the provided feedback, we have uploaded our updated paper with the supplemental material mentioned and other minor changes.
> > >
> > > Again, thank you for the review and insightful comments!

---

> > > > ### Comment · Reviewer_pmf6 · 2022-08-09
> > > > **RE: Subsequent Respone**
> > > >
> > > > Thanks for the clarification and elaboration!  I appreciate it.

---

> ### Author Response · Authors · 2022-08-07
> **Author Follow Up**
>
> As the discussion period is coming to an end, we wanted to thank you again for the review and suggestions. If there are any questions remaining or areas we can help clarify, please let us know.

---

### Meta-Review · Area_Chair_NkAy · 2022-08-26

**Recommendation:** Accept
**Confidence:** Less certain

**Metareview:**

The paper considers a collaborative decision-making setting in which one agent can suggest actions for a listening agent to execute. The listening agent is not bound to take these suggested actions. The paper models the listening agent's decision-making process as a POMDP, treating the suggestions provided by other agents as observations (i.e., dependent only on the state) that are used to update the belief state. The paper describes two representations of the suggested actions and presents empirical results on simulated domains that demonstrate that the listening agent's performance improves when following the suggestions of a perfect oracle, while it degrades given imperfect suggestions in proportion to the level of noise.

The paper was reviewed by three researchers who read the author response and discussed the paper together with the AC. The reviewers largely agree that the collaborative decision-making problem as formulated is interesting. The reviewers find that the idea of formulating suggestion-following as a POMDP with suggestions modeled as observations is clever, and that the description of the proposed formulation is clear and easy to follow. The two formulations of the observation function are reasonable, albeit very simple initial models, though the paper does not provide much insight into when one should be used over the other. A primary concern with the paper is that it only provides empirical results. The work would have significantly benefited from theoretical results, which would help to understand how the method would generalize to other domains.

**Award:**

No

---

### Decision · Program_Chairs · 2022-09-14

Accept